# Digital Identities of Young People from the South of Spain: An Online Sexual Differentiation

**DOI:** 10.3390/children8121179

**Published:** 2021-12-14

**Authors:** E. Begoña García-Navarro, Jose Luis Gil Bermejo, Miriam Araujo-Hernández

**Affiliations:** 1Department of Nursing, University of Huelva, 21007 Huelva, Spain; miriam.araujo@denf.uhu.es; 2Research Group Social Studies and Social Intervention, University of Huelva, 21007 Huelva, Spain; 3Department of Sociology, Social Work and Public Health, University of Huelva, 21007 Huelva, Spain; gestalt.jose@gmail.com

**Keywords:** adolescents, children, social networks, identity, gender inequality, gender digital divide

## Abstract

Through a mixed methodological approach, we want to know how adolescents aged between 14 and 16 years from the south of Spain express and identify themselves on social networks, with respect to their sex. As such differences can determine gender inequality, we will analyse differences between females and males regarding the expression of identity on social networks. Analysis of obtained results demonstrates that many relevant attributes still emerge such as the socio-cultural representation of gender as sex in social networks. Differences emerged between the identity expressions of females and males which can generate inequalities favouring females and males. This implies a series of repercussions and, ultimately, defines the so-called digital gender divide. Taking into account these results we could intervene in the population of children to carry out prevention activities focused on social networks.

## 1. Introduction

This article presents research carried out in the south of Spain, specifically in the province of Huelva, characterised by a social context that is both rural and urban. This represents a special geographical context, offering a diversity and abundancy of unique discourses and realities within the Spanish geographical framework. On the one hand, this context represents a society characterised by modernisation but with traits of localism that are more related with rural environments. On the other hand, this context presents a society characterised by post-modernisation, sharing the idiosyncrasies or characteristics of globalised urban geographical hotspots [1].

The examination of teenagers and the use of social networks is and will be a matter of great relevance, in relation to both current daily realities and the future [2]. At present, social networks are tailored representations of our sociocultural system, in that they transmit and legitamise patriarchal hegemonic values [3]. Due to the prominence that social networks have in our society, particularly in adolescence [4], the present research strives to deepen knowledge from this perspective. This research will also take a gender perspective, specifically considering gender as a binary and limiting construct represented by males and females. In this way, it is possible to observe how females and males identify themselves and each other, to what extent they differ, their shared similarities, and to what extent patriarchal values are reproduced regarding feminine and masculine identities [5].

When we refer to the daily performative expression of young people on social networks, we consider the generalised expression of thoughts, cognitions and actions [6] that are chemically modified according to sex and gender. In this sense, actions may occur that produce effects and consequences on other actions, whilst also producing repetitive, ritualised and unoriginal acts associated with our established culture and norms.

The identity of the self as a gender identity with all its attributions and elements, must be understood as an expression of internal and external dynamics of the person [7]. However, it must not be forgotten that the individual is an active agent in the construction of their own identity [8]. On the other hand, gender represents a type of performativity in itself with respect to sex. Despite the differentiation of these two concepts [9], we are aware of the strong cultural association that exists between them and how difficult it is, therefore, to establish models or performativities that are different to those already established, in regards to both femininity and masculinity.

Paradoxically, the dual gender division between females and males does not have space to accommodate current postmodernist outlooks in which borderline sexual identities appear through dissident subjective discourses, such as queer theory, and make us question the dichotomous polarity of both sexes [10]. At present other debates are still open such as those that delve deeper into what it is to be queer. Such debate is reflected through theory, policy and examination of daily practice, incorporating aspects such as intersectionality, gender disobedience or queer transfeminism [11]. Such perspectives give broader gender considerations a voice. However, they open up the possibility for new forms of diversity and practice, with the search for alternatives having a concomitant cost.

It seems that the daily reality of adolescents’ social network use in western countries, is moving away from the ideal—engaging “online” with the social network—to the real—engaging without being connected to the internet or “offline”.

According to the National Institute of Statistics (INE), 66% of young Spaniards between 10 and 15 years old already own a mobile phone. A total of 88.14% of young people surveyed in the I Observatory of Generation Z used their mobile for leisure, to chat on WhatsApp or play [12], that is, in the last stage of childhood, children begin to make use of tools that open the doors to a digital world.

Offline realities are becoming more and more diffuse, becoming integrated alongside online realities into the concept of the self. Thus, it is undergoing a rapid process of constant socialisation and subjectivation [13] within the adolescent identity. Social networks are used to express and create an “ideal self” [14]. Our ideal self is represented as an ephemeral, superficial and sometimes unattainable identification of oneself. In it, one it socially accepted within a given culture, subculture or counterculture [15].

Gender stereotypes foster the differential socialisation of girls and boys, progressively influencing their cognitive and social development [5,9], through the media, and more specifically through social networks [12], as we will discuss in this paper.

In today’s society, educational proposals are increasing that bet on an early prevention of abuse or misuse of social networks in the child population [16], in this case it is fundamental, since an education that promotes gender equality in the use of social networks is very positive, if we want to propose a more egalitarian society in the future [17], in this line we consider as necessary the development of this research. We need to delve specifically into whether women and men show a differentiated and stereotyped use of social media, as this represents the heart of patriarchal dualism [18].

We question whether a digital gender divide has emerged that increasingly—and from early ages—differentiates women from men, leading to inequality that magnifies andocentrism and negatively affects women.

## 2. Materials and Methods

The aim of the present work was to examine how young people in a municipality in southern Spain—Huelva—identify themselves in social networks. In this way, we questioned whether this identification corresponds to established and differentiated gender patterns that are characteristic of girls and boys.

This work followed a quantitative and qualitative methodological approach based on the standpoint of methodological pluralism. It is supported by data triangulation [19], with the combination of information increasing coherence between research methods and the object of study. A mixed methodology is hugely appropriateness and of great relevance given the need of achieving greater and more profound understanding within the field in order to address subtle issues related to gender differences and inequality [9].

An observational design was developed which was descriptive and cross-sectional in nature. The quantitative sample was selected through non-random convenience sampling. This was made possible sue to the ease of access granted by the participating educational centres in the province of Huelva. When selecting educational centres, the number of inhabitants in the locality was taken as a reference. Urban and rural settings were defined as having more or less than 5000 inhabitants, respectively. We selected a total of 7 public educational centres, with 4 coming from urban settings and 3 from rural settings.

### 2.1. Participants

For both the quantitative and qualitative sample inclusion criteria stipulated that participants must be aged between 14 and 16 years and be enrolled as students at public educational centres in non-urban/rural environments (towns with less than 5000 inhabitants) or urban environments (towns with more than 5000 inhabitants) in the province of Huelva. At a quantitative level, the sample was made up of 400 young people (200 girls and 200 boys) who completed all questionnaire items (see Table 1). All participants were students undertaking compulsory secondary education (ESO) and came from 7 different public schools. The average age of participants was 15.01 years (SD = 0.82).

Regarding the selection of informants for the qualitative sample, it was carried out by means of an intentional non-probabilistic sampling, contemplating young people between 14 and 16 years old with the inclusion criteria described above (taking into account the rural and urban environment).

For the selection of interviews and focus groups, a snowball sampling method was used. Thus, the total number of interviews was not determined a priori but from a process of theoretical saturation [20], a criterion widely used in non-probabilistic sampling, obtaining a total of 33 interviews.

With the selected population, three discussion groups were formed, one consisted of 8 girls, another was formed by 7 boys and a third was a mixed group of 10 students. Further, 8 in-depth interviews were conducted, four with girls and four with boys. Interview participants were different to those who had attended the discussion groups. The qualitative sample, therefore, comprised a total of 33 students from urban and rural areas.

The choice of these groups was driven solely by the search for actors of both sexes, who voluntarily wanted to participate and were users of social networks.

### 2.2. Instruments

The following instruments were used for data collection:-Questions of a socio-demographic nature: age, sex and area of residence (rural or urban);-Sex role inventory [21]: This considered the attributes of competitiveness, leadership, aggression, tenderness, independence, femininity, masculinity, jealousness and individualism. This was administered alongside the instrument described in a recent study of the Spanish adolescent population: “Strong as a father? Sensitive as a mother? Gender identities in adolescent youth” [22]. This is a nationwide survey study of 2154 schooled young people aged between 14 and 19 years, and considers the following attributes: sensitivity, image concerns, responsibleness, intelligence, dependence, home-loving, autonomy, superficialness, possessiveness/jealousness, independence and romanticism.-In relation to social networks, participants are asked about their motives for using social networks and to identify the networks they most commonly used. This item was multiple choice, with the following options provided: Twitter, Facebook, YouTube, Instagram, Snapchat, blogs, and other social networks including WhatsApp, although this is not a social network itself.

Satisfactory outcomes were obtained for the variables that make up this instrument, with adequate internal validity (α = 0.82).

The script of the in-depth interview as well as the discussion groups begins with the preliminary categories: differentiated use of social networks, references to the type of social networks and reasons for their use, and attributes associated with the identification of either of the genders. Subsequently, emerging categories emerged as the fieldwork was carried out.

### 2.3. Procedure

Once access was granted from the educational centres, the day and time for carrying out the questionnaires, interviews and discussion groups was agreed upon and all educational centres were sent an informed consent form. In the case that students agreed to participate in the research, the form was signed by their parents or legal guardians. The study guarantees data confidentiality and participant anonymity. Participants were well informed about the study objectives and the bioethical principles of the Declaration of Helsinki were always respected. Likewise, data handling in the research complied with current national and international regulations regarding the protection of personal data. The project was approved by the Ethics Committee of the Council of Andalucía.

All participants were undertaking the educational level corresponding to the 3rd and 4th years of ESO. The day the questionnaire was carried out in the classroom, discussion groups were organised and interviews were held with students who had voluntarily agreed to participate in one of these two qualitative procedures. The described qualitative techniques were then carried out the following week.

### 2.4. Analysis

All quantitative analyses were carried out using the statistical analysis program SPSS V.25, with all basic variables being compared (*n* = 400).

The qualitative data processing program ATLAS. Ti. V8 was used. Interviews were recorded and transcribed. The research team listened to and read the interviews in order to make an initial superficial interpretation. This provided a general idea which supported a more in-depth analysis (identification of relevant recurring themes, search for similarities and differences between themes in order to develop codes–dimensions and, with these, thematic categories. The repetition of codes–dimensions on behalf of researchers—blind analysis indicated that the analysis captured the essence and exposed the meaning of the studied phenomenon).

The following categories resulted from this analysis: differential use of social networks, types of social networks used and for what purpose, and the attributes associated with gender identification (see Table 2).

## 3. Results

In order to investigate the performative expression of adolescents in social networks, we first sought to uncover the social networks that are most frequently used by both girls and boys, alongside the reasons for their use. The first step was to observe whether differences already existed between the general uses of these networks and the way in which adolescents are defined and represented by social networks. We will first present the results obtained at a quantitative level, followed by a series of qualitative findings relating to each of the aforementioned mentioned questions.

### 3.1. Use of Social Networks by Females and Males

From the perspective of the digital gender divide, there is no differential use of social networks between females and males. Facebook, followed by Instagram, Whatsapp and YouTube were the social networks or applications most frequently used by youth. According to Chi-square analysis (χ^2^ = 17.26; df = 1; *p* < 0.001), significant differences were observed in blog use. In this case, females (85%) used blogs to a greater extent than males (65.7%). Without a doubt, blogs are more reproducible in nature and are more durable over time. They are increasingly more personal and intimate spaces relative to other social networks spaces or tools. Interestingly, these aspects appeal more to females, as depicted by the discourses produced through the previously described qualitative analysis (Table 3).

We found differences in the use of social networks. There was a trend towards a preference for direct and instantaneous diffusion of material on social networks such as WhatsApp or Instagram relative to blogs or Youtube. More specifically, in urban environments the former, faster modality was preferred, whereas in rural environments the latter, slower modality was preferred. This aspect emerged through qualitative analysis, since quantitative analysis did not show any significant difference between rural and urban environments.

However, although there are almost no differences between the social networks most used by youth, nuances can be seen in the motives for using them and the way in which they are used. This can be seen in the outcomes pertaining to the multiple-choice question of ‘what do you use social networks for?’ (see Figure 1). To some extent, social networks are used by both girls (93%) and boys (86%) to talk with friends. Next, the second main motive for boys is to talk with family (74%), whereas in girls it is to talk with their partner (69%). Statistically significant differences are seen between boys and girls in the use of social networks for the undoubtedly more competitive activity of flirting (χ^2^ = 25.07; df = 1; *p* < 0.001), with 62.8% of boys stating this reason relative to 35.8% of girls.

This information is confirmed in the analysis of the discourses of the population studied, which manifests this difference in the motivation of the use of networks. Below we can see some expressions (Table 4.)

When writing in social networks, girls maintain more intimate conversations relative to boys, with the latter steering away from the reproductive in favour of the productive. In the discussion group, the topic of homosexuality was broached in both female and male groups, with females even opening up with regards to male homosexuality. It was observed that males are more inclined to use concrete terms such as to meet, to play or to watch videos. They tended to use social networks as an end in itself, in a more utilitarian or outcome-oriented way. Females blend in more personal aspects related to their intimate life, personal life and their vicarious experiences. Another aspect to emphasise is the importance of the mobile phone as a tool through which young people mostly access social networks.

An aspect that has already been touched on at a quantitative level also emerged from the content analysis. This pertained to a new use of social networks identified by both females and males and related to the use of social networks to engage in gossip about others (Table 5.):

It seems that girls and boys from rural environments describe social network use for gossip as something habitual. It is even quite common for individuals to mention that they go as far as adding false profiles to hide their identity and follow people without them knowing that they are being talked about. This is, without a doubt an interesting use of social networks as it invades the privacy of others whilst protecting the identity of potential perpetrators of gossip.

Although there are no great differences in the social networks used by boys and girls, differences start to be seen in how they are used and the content that is uploaded to them. The qualitative analysis conducted here uncovered this important nuance.

### 3.2. How Adolescents Are Defined on Social Networks

Quantitative intra-group analysis comparing the group of girls and the group of boys with regards to their perceptions about how they themselves are considered on social networks obtained significant differences for most categories (see Table 2). For instance, in relation to sensitivity (χ^2^ = 64.64; df = 1; *p* < 0.001), females (80.4%) consider themselves to be more sensitive on social networks than males (37.8%), likewise females (71.7%) are more concerned about their image than males (54.1%), (χ^2^ = 11.46; df = 1; *p* < 0.001). As demonstrated through discourse, it should be kept in mind that boys are increasingly concerned about their image, although trends remain towards gender differences relating to image perceptions.

The continuum between tenderness and aggressiveness also reveals the two genders to be polar opposites with gender differences being reliably replicated throughout the study processes. On the one hand, tenderness (χ^2^ = 12.26; df = 1; *p* < 0.001) is more strongly linked to the feminine identity (61.9%) as opposed to that of boys (43%). The opposite occurs with aggressiveness (χ^2^ = 22.89; gl = 1; *p* < 0.001), where boys (55.8%) obtain a higher percentage than girls (30.6%).

Another role closely associated with males is leadership. It can be seen that 54% of boys defined themselves as leaders, compared with just 34% of girls (χ^2^ = 63.57; df = 2; *p* < 0.001). This aspect is strongly related with patriarchal influences, with males tending to assume roles and attitudes of dominance or command in life.

Further, a large difference was found according to sex (χ^2^ = 84.22; df = 1; *p* < 0.001) with regards to perceptions of competitiveness, with boys (80.7%) more strongly identifying themselves as competitive than girls (31.8%). This is unsurprising given that relevant attributes are strongly associated with masculinity.

Another aspect to be strongly associated with males was the perception of being independent (χ^2^ = 20.26; df = 1; *p* < 0.001), with this trait being less apparent in girls (54.2%) than boys (77.2%). A similar pattern was seen in relation to autonomy, with differences again emerging between girls (44.4%) and boys (90.1%) which were even more significant in this case (χ^2^ = 83.93; df = 1; *p* < 0.001). In contrast, we can see that the trait of dependence (χ^2^ = 27.49; df = 1; *p* < 0.001) was also inversely related with the previously discussed characteristic. Dependence was seen to more strongly emerge within females (49.1%) than males (22.1%).

Masculinity was the characteristic for which more differences existed (χ^2^ = 150.18; df = 1; *p* < 0.001) between females (13.3%) and males (79.1%). Interestingly, when comparing these outcomes with those pertaining to perceptions of femininity (females: 68.1%, versus males: 10.5%) there is again a significant difference between both groups (χ^2^ = 120.42; df = 1; *p* < 0.01) but it is clear that girls are more able to identify with the masculine pole than boys are with the feminine. This may be evidence of androcentrism already being present at these ages.

Perceptions of whether one identifies as being a homebody showed large differences between girls and boys (χ^2^ = 19.97; df = 1; *p* < 0.001). A total of 61.9% of girls identified with this trait compared with 37.8% of boys. This suggests that, even at these early ages, this attribute already appears in accordance with the roles likely to be played in both public and private spheres. It is of huge interest the way in which the home, as a private and reproductive context, takes on such relevance in women from as early as 14 to 16 years old.

Other data did not show statistically significant differences in relation to the reality under investigation (see Table 6). Some of these outcomes are equally concerning, for instance those pertaining to jealousness or possessiveness. Both genders show high levels of identification with these traits on social networks, despite there being no significant differences between girls (53.2%) and boys (63.4%). This means we are currently faced with a situation in which a high percentage of young people identify with a trait that endangers inequality in relation to dominance or control over another other person, in this case in the context of a loving partner.

Quantitative data produced results that forcefully demonstrate that commonly established gender identities are being reliably reproduced when young people express themselves on social networks. When considering the characteristics examined, we can observe that females self-identify more strongly with sensitivity. This sensitivity was not shown by males who were more concerned with more direct or superficial issues. A relationship also emerged in which sensitivity was more relevant in rural environments, whereas image was more relevant in urban environments. Interestingly, boys did not only state this when referring to girls, they also recognised that such traits pertained to themselves:

Moving on to consider what young people express on social networks (Table 7), we will now introduce two categories to emerge from the discourse. Namely, these categories are photographic images and emotions. A number of observed variables relating to these two categories were seen to be of great relevance to the qualitative analysis.

Beginning with image (Table 8), we observed that females were more likely than males to upload images relating to their body and aesthetic beauty. This is a differential aspect which could have repercussions, particularly in relation to intimacy, with females being in a situation of vulnerability in relation to males. We can also observe that image is more likely to emerge within urban women.

There is a feeling of self-control and freedom in the discourse of the women who express their story in this way. They are little critical of the fact that they post images, even when they are controversial or intimate. They show a desire to be seen on social networks and speak out against aggressions that they may have been exposed to by other females or by males. On the other hand, there is a critical and conscious outlook pertaining to the difference between sexes in the degree of intimacy shown in the photographs exhibited on social networks, in addition to considerations of their risk for girls (Table 9):

It seems that the image (Table 10) shown is another expression of the demands made by boys themselves, although demands can also be placed on them since many boys also give importance to their image. On the other hand, girls also vindicate their image to other girls as well as boys. This exposes them to other alternatives to heteronormativity and indicates a self-affirmation of the need to be beautiful for oneself more than in the view of another person. As we previously observed, images have a potential destination, normally another person. In the accounts given by urban girls and boys no concrete reference was given. Images are, therefore, more and issue of making a social demonstration, as shown in the following accounts:

In the previous section we examined image, in its social, personal and private essence, now we will delve a little deeper into something with similar characteristics but more intimate connotations. Specifically, we will now consider the level of emotional expression that occurs on social networks.

As for the content or use of social networks both girls and boys recognise that emotional expression is more typical of girls than boys (Table 11). This is demonstrated through the following accounts:

When it comes to expressing feelings, there seems to be a tendency for men to associate emotional expression with a partner with weakness. They see it as moving away from the stereotype of masculinity given the belief that the masculine part is the part that does not suffer, does not cry and is more autonomous. With regards to geographical differences, examination of the accounts reveals a discourse that leans more towards concrete ideas in rural populations and more abstract ideas in urban populations.

Further, it was observed that certain emotions (Table 12), expressed in a more intimate sense by girls, are preferentially posted in more private or less diffused media using applications such as WhatsApp. In contrast, more social or diffused applications are used to show cheerful emotions.

This aspect of emotional display is associated with image and a gender difference is again seen, with girls expressing emotions more than boys. Girls also express emotions to a greater extent on social networks when those emotions are positive, and through less widespread social networks when they are negative.

Given this marked digital gender divide, we can observe that girls express more identarian feelings of friendship between themselves from a place of understanding and care. In this context, sharing feelings and intimate information makes a lot of sense. Where social networks are put into play the private is made public to some extent, where the private involves intimate details given in an emotional context. Relative to males, this again shows a greater willingness of females to expose private and intimate aspects, despite the various consequences that this may have with respect to greater vulnerability to conflict situations.

Next, we will show a table (Table 13) where the main results are described based on the objectives set out in this study.

## 4. Discussion

Results show that there are no significant differences between the social networks used by young people. For both females and males Facebook, followed by Instagram, Whatsapp (as an instant messaging service) and YouTube were the most used social networks or applications. This finding is corroborated by international studies [23,24]. This shows how widespread the use of social networks is amongst adolescents, irrespective of sex, with the more relevant aspect for young people [25] being access to mobile phone services.

In this sense, the way in which each young person presents themself or identifies with themself on social networks [5] is hugely important. It is here that the so-called gender divide has been emerged, with differences being seen between girls and boys. The image presented on social networks has been shown by the present research to be a differentiating aspect, although it is true that males are showing growing concern for their body image [26]. Even so, females continue to expose themselves and put themselves in vulnerable positions to a greater extent on the different social networks, in order to achieve an image that meets societal beauty ideals [27]. On the other hand, the use of Selfies is increasingly related to a desire to build a public reputation [28] throughout the narcissistic construction of identity during adolescence [29]. Posted images are still aspects of private life that remain unprotected due to the open and almost unlimited access given in virtual communities. Another aspect of relevance, which also influences privacy and intimacy, is that the digital gap emerged in the general expression of feelings and emotions on social networks [30]. In this sense, females are shown to be more expressive than men who, in turn, are more instrumental in their use of social networks. Again, this issue may impact upon the greater vulnerability of females with respect to males due to the fact that they include more private and personal aspects in their posts to different social networks.

At a national level, studies related to gender stereotypes in Spain show us how image, ideals and stereotypical images of men and women remain static [22]. This is despite social changes such as the influence of the feminist movement and LGTBIQ+ policies, which encourage a more open view of heterocentric binarism from which gender can be reconceptualised and separated from the classic cisgeneric vision that has culturally marked the identities of men and women in Spain.

If we extrapolate to social networks, the same conclusions can be made [31]. At an international level the differential association of identity with the masculine or feminine gender seen in girls and boys on social networks is being increasingly considered authors [32].

Jealousness, which emerged in both collected data and discourse, was commonly expressed by girls and boys. This is an undoubtedly worrying aspect as it will likely lead to affective relationships that are marked by possessiveness and aggressiveness [33]. This is an aspect that is increasingly present in adolescence and, especially, in adolescent’s use of social networks [34,35].

In urban females we observe that in certain situations that involve compromising images, there is a belief that one has control over the situation [36]. In contrast, in rural areas females feel a greater influence of social control with regards to stepping out of established patterns [37].

In urban females it was observed that, despite differences in gender roles, there is a critical standpoint that involves the questioning of and resistance to the regulatory environment and cultural norms established by the post-modern androcentric vision [38,39,40]. New expressions of gender were shown, alongside alternatives for new egalitarian gender approaches on social networks [41,42].

With the results that emerge from our research, educational policies that bet on the inclusion of the use of social networks as a curricular subject in school at an early age could be reinforced. We should not only work for students, but also for the people who make up the educational community [17,43,44], since the gender models established in social networks are a reflection of the reality lived in childhood [16,43]. It would also be interesting to work with the family in order to encourage a healthier contact in times, contents, functions and widespread use of social networks.

## 5. Study Limitations

In this research topic, it would have been very relevant to approach how the different male and female roles develop in homosexual adolescents, both in men and women [45,46] and to be able to investigate whether women reproduce female stereotypes and men, beyond their sexual orientation. Another relevant aspect, which as a limitation of this study we point out, is the analysis of situations of harassment through social networks or cyberbullying, which, although we have investigated a little on the subject, it would be necessary to influence the typologies of aggressions that can be generated both girls and boys [47,48,49], on the other hand, as indicated in the study, it is of great importance to continue betting on educational programs for the prevention of cyberbullying [50].

## 6. Conclusions

By posing the question relating to whether differences exist between girls and boys on social networks with regards to their identification with gender roles, it was observed that gender stereotypes persist within both sexes with certain important nuances.

Although there is apparent equality in the use of social networks, with girls and boys using the same social networks and for similar reasons, a dichotomy emerged between the instrumental representations of boys and the expressive representations of girls. Males were more likely to use social networks in a concrete way, to meet friends, flirt, send something, see information, play, etc. Private content is less often exposed publicly by males, who instead post public content to private spaces. In contrast, females make their private space public through emotional expression and are more likely to generate bonds through relational communication.

Regarding image exposure (photographs and videos of oneself (selfies) or with others), more social pressure is felt by females to be seen in videos or photographs, especially in rural areas. In general, a more sexualised model appears as a representative figure of beauty according to the culturally established heteropatriarchal ideals.

The relationship between the productive and the reproductive, or the instrumental and expressive, between girls and boys, respectively, leads young people to relate with their established gender roles.

In both girls and boys, a highly concerning normalisation of jealousness/possessiveness appears. This behavioral pattern is reinforced through social networks, via the loss of control and multiple forms of aggression that take place.

We can say that girls show more intimate expression on social networks than boys. This is seen through the expression of feelings and the sharing of pictures with more sensual and sexualised content. This greatly compromises textual or image privacy within social networks, promoting greater psychological and social vulnerability of girls than boys. Given this finding, we assume that young people have little or no awareness of the existence of a worrying digital gender gap.

It would be interesting for future research to take a ruptured vision of heteronormativity and address more trans-gender aspects, in addition to new femininities and masculinities. On the other hand, we consider it a limitation of the present research that it did not delve deeper into perceptions of control amongst females who purport to be knowledgeable of the risk they sometimes take when exposing content on social networks.

In contrast to the dual reality of girls and boys in social networks, new forms of expression and identification are observed on social networks, especially amongst women. Such expressions are related with already established cultural elements which will enrichen and blur the gender duality in present and even future post-modern societies.

## Figures and Tables

**Figure 1 children-08-01179-f001:**
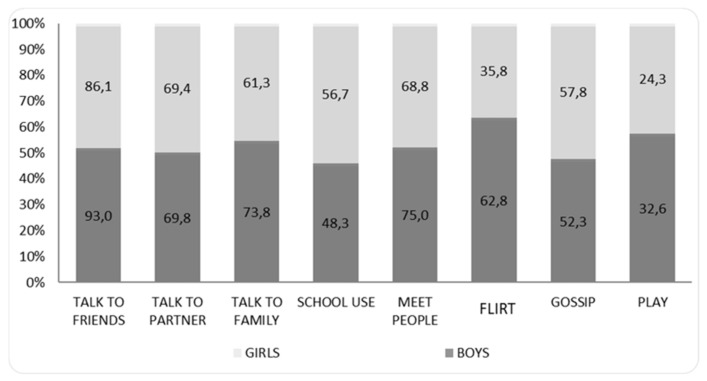
Use of social networks according to sex. Results come from responses to the question: ‘What do you use social networks for?’.

**Table 1 children-08-01179-t001:** Sample: age, sex, and geographical location.

Categories	Geographical Location		Total
	Urban	Rural	
**14 years**	100	32	132
**15 years**	96	38	134
**16 years**	85	49	134
**Men**	144	56	200
**Women**	137	63	200

**Source**: developed by the author.

**Table 2 children-08-01179-t002:** Categories and subcategories to emerge from the study according to different techniques.

Dimension	Categories	Subcategories	Interviews	Discussion Groups
** *Identification on social networks* **	Use of social networks by females and males	Productive use = guys Productive use = girls	x	x
		Facebook, WhatsApp, Instagram = guys Blogs, Youtube = girls	x	x
	How adolescents are defined on social networks	Sending photographs and videos	x	x
		Talking about feelings		x

**Source**: elaborated by the authors in relation to data obtained through qualitative analysis. **NOTES** 1. Coding for category transcription: [UF16]: 16-year-old urban female. [RM15]: 15-year-old rural male.

**Table 3 children-08-01179-t003:** Use of personal and intimate spaces in relation to other spaces or social media tools.

Categories	Subcategories	Geographical Location and Sex	Disocurses	
Use of social networks by males and females	Facebook, WhatsApp, Instagram = guys Blogs, Youtube = girls	Urban Female	UF16	We are more about telling each other through WhatsApp, between us or writing to our boyfriend, and of generally posting on Instagram or Facebook, everything goes on there, ah and of giving little away [UF16]
			UF15	I like that I end up relating to the people who connect with my stuff, having similar tastes [UF15]
			UF14	I prefer blogs or YouTube channels, I learn about things I like [UF14]
		Rural Female	RF16	In a blog you don’t have the problem that somebody is there who doesn’t want to be, it gives more privacy than other social networks [RF16]
		Rural Male	RM15	I do use Facebook and WhatsApp a lot, I like to be seen and to see others, and for this you always have to be aware of who the audience is and who is publishing something of interest to you, its very easy on your mobile [phone] [RM15] I don’t usually entertain myself at home, I like to go out, I am less interested in social networks, spaces where you have to write, I am more about Instagram and WhatsApp [RM15]

**Table 4 children-08-01179-t004:** Motives for using them and the way in which they are used girls and guys.

Categories	Subcategories	Geographical Location and Sex	Disocurses	
Use of social networks by males and females	Productive use = guys Productive use = girls	Rural Male	RM14	Girls often set up a false profile to see what other girls are doing, girls who are not their friends and for whatever reason they do not send a friend request [RM14]
			RM16	I have sometimes thought of looking at other girls to see what they are wearing or who they are hanging out with, I don’t know if I know more about them, it’s a way of getting closer to someone you like or whatever and them not discovering that you’re following them [RM16]
		Rural Female	RF14	On Facebook and Instagram you hear a lot about fake profiles to gossip about people, it’s a good way to find out whatever you want while nobody knows what you’re looking at [RF14]
			RF15	The gossip is more our interest, the younger ones don’t care about that stuff [RF15]

**Table 5 children-08-01179-t005:** Use of social networks to engage in gossip. Different Female, Male.

Categories	Subcategories	Geographical Location and Sex	Disocurses	
Use of social networks by males and females	Productive use = guys Productive use= girls	Rural Male	RM14	Girls often set up a false profile to see what other girls are doing, girls who are not their friends and for whatever reason they do not send a friend request [RM14]
			RM16	I have sometimes thought of looking at other girls to see what they are wearing or who they are hanging out with, I don’t know if I know more about them, it’s a way of getting closer to someone you like or whatever and them not discovering that you’re following them [RM16]
		Rural Female	RF14	On Facebook and Instagram you hear a lot about fake profiles to gossip about people, it’s a good way to find out whatever you want while nobody knows what you’re looking at [RF14]
			RF15	The gossip is more our interest, the younger ones don’t care about that stuff [RF15]

**Table 6 children-08-01179-t006:** Identification with defined traits on social networks.

TRAIT	SEX	Intra-Group Percentages (Girls and Girls/Boys and Boys)	Inter-Group Percentages (Girls and Boys)
Sensitivity ***	girls	80.4%	31.9%
	boys	37.8%	68.1%
Image awareness ***	girls	71.7%	57.1%
	boys	54.1%	42.9%
Responsibleness **	girls	74.5%	54.7%
	boys	62.2%	45.3%
Individualism ***	girls	22.5%	31.7%
	boys	48.7%	68.3%
Tenderness ***	girls	61.9%	59.1%
	boys	43%	40.9%
Intelligence	girls	65.3%	49.8%
	boys	66.3%	50.2%
Dependence ***	girls	49.1%	69.1%
	boys	22.1%	30.9%
Femininity ***	girls	68.1%	86.8%
	boys	10.5%	13.2%
Masculinity ***	girls	13.3%	14.5%
	boys	79.1%	85.5%
Autonomy ***	girls	44.4%	33.1%
	boys	90.1%	66.9%
Superficialness ***	girls	30.6%	33.2%
	boys	62.2%	66.8%
Jealousness/Possessiveness **	girls	53.2%	45.7%
	boys	63.4%	54.3%
Independence ***	girls	54.2%	41.4%
	boys	77.2%	58.6%
Competitiveness ***	girls	31.8%	28.4%
	boys	80.7%	71.6%
Homeliness ***	girls	61.9%	62.2%
	boys	37.8%	37.8%
Romanticism ***	girls	68.8%	61.1%
	boys	44.2%	38.9%
Leadership ***	girls	34%	26.45%
	boys	54%	73.55%
Aggressiveness ***	girls	30.6%	35.6%
	boys	55.8%	64.4%

Note: ** *p* < 0.05, *** *p* < 0.001.

**Table 7 children-08-01179-t007:** Expression of feelings in social networks.

Categories	Subcategories	Geographical Location and Sex	Disocurses	
Use of social networks by males and females	Productive use = guys Productive use = girls	Rural Male	RM15	Girls are more about sharing their sorrows or joys, we get more to the point, and when we tell something we don’t make it very public and tell it to one or two friends [RM15] I don’t want them to see me suffer, whilst they could share my joys, I already celebrate that with my friends, but a photo shows how happy I am and makes me happy [RM15]
			RM16	We prefer to tell our things to close friends, in networks or share in private WhatsApp groups, we don’t send photos so much [RM16]

**Table 8 children-08-01179-t008:** Images relating to their body and aesthetic beauty.

Categories	Subcategories	Geographical Location and Sex	Disocurses	
Use of social networks by males and females	Productive use = guys Productive use = girls	Urban Female	UF14	Yes, we care more about image in the photographs we post, maybe because of what they might say, about our image, and you notice that on the networks, we don’t just post anything [UF14]
			UF15	As girls we show more, but only what we want, sometimes we insinuate, but we have control in the background [UF15]
			UF16	Sometimes amongst ourselves we attack each other when somebody shows more than they should, or sometimes it is the boys who insult you, other times the outcome is better and many comments appear and you can see that people have seen you and liked the photo you posted [UF16]
		Urban Male	UM16	We don’t really ever post photos or anything, or sometimes when we are partying but less of us ourselves, we like to be seen to be having a good time [UM16]

**Table 9 children-08-01179-t009:** Difference between the sexes in the degree of intimacy shown by the photographs displayed on social networks.

Categories	Subcategories	Geographical Location and Sex	Disocurses	
Use of social networks by males and females	Productive use = guys Productive use = girls	Rural Female	RF16	Its not only for them that we get pretty, we can also have relationships between us girls, its not just about being a couple, we also like to be pretty for other girls [RF16]
		Urban Female	UF16	Well, sometimes you send photos to attract attention, usually from people you like, to flirt, sometimes I admit that I’ve gone too far, but for now I’ve been lucky and have not had any negative repercussions from them [UF16]
		Urban Male	UM16	The image isn’t so important, man, let’s see, you’re not going to look a mess, but if you show muscle that’s cool, but maybe you don’t have to show the face or how you’re dressed, that doesn’t matter, you can go crappy and nothing happens [UM16]

**Table 10 children-08-01179-t010:** Body image.

Categories	Subcategories	Geographical Location and Sex	Disocurses	
Use of social networks by males and females	Productive use = guys Productive use = girls	Rural Male	RM14	To see the guys we dress one way and for the girls well each to their own, that does not mean that we go looking a mess, if we want the girls to notice us we have to look good, like good boys we know how to how to make them like us and who to make like us [RM14]
			RM15	Well, if you work on your body, like with muscles and then you know how to look good in clothes that is key to making the best of what you’ve got, with the girls, with whoever it is you meet I’m a boy, I do not say whether I like girls or boys, I like to look handsome and for them to see me [RM15] Above all I take selfies with my friends, I like to see myself with them and know they are there [RM15]
		Urban Female	UF16	A good selfie saves your life, when you’re down you look at it, send it to your pals, or just to flirt with someone in a chill sense [UF16]
		Urban Male	UM14	As a boy I consider what I wear, how I appear in photos and how I carry myself, looking good doesn’t have to be just for girls, it’s also for us [UM14]
			UM15	There is a lot of nonsense with selfies, I usually do them alone so I give a more serious and mysterious image of myself, but then nobody takes me seriously [UM15]

**Table 11 children-08-01179-t011:** Emotional expression on social networks.

Categories	Subcategories	Geographical Location and Sex	Disocurses	
Use of social networks by males and females	Productive use = guys Productive use = girls	Rural Male	RM14	Boys are not really into talking about our things, about our problems, and less about girls, that would be ridiculous [RM14]
			RM15	It’s embarrasing, to tell a girl your issues, even if it is your friend or your partner, it puts you in a bad position, then you learn that they are saying things about you, I know people who this has happened to [RM15] When I want to flirt and pull out the stops I tell them about me, in a sensitive way, well sometimes, you more or less know what they like [RM15]
		Rural Female	RF16	Boys are mute, you ask them how they are, if they have something going on or anything, and they don’t answer you, they stay quiet, it seems that nothing ever happens to them, but that’s not the case [RF16]
		Urban Female	UF16	But we like that the guys also share with their issues with us, and more if we are in a relationship, in the way that we girls do, and that doesn’t make them lesser men [UF16]
		Urban Male	UM14	We like send photos more, with friends or to meet up, but not to write, from time to time maybe something a bit affectionate, but to look good [UM14]

**Table 12 children-08-01179-t012:** Expressing feelings on social network.

Categories	Subcategories	Geographical Location and Sex	Disocurses	
Use of social networks by males and females	Productive use = guys Productive use = girls	Rural Female	RF15	In networks I don’t usually post many things that happen to me, that’s why I use WhatsApp, it’s more intimate, especially with friends [RF15]
		Urban Female	UF16	Always keep in mind the people who may be seeing you, it is better to talk about your things in private, especially if you end up getting it wrong, I do not want people to see me get it wrong [UF16] When I want to share joy or good things that happen to me I do that on Facebook or Instagram, with a cool photo, which is there forever and everyone knows, now when I’m sad I cut myself off a little more [UF16]

**Table 13 children-08-01179-t013:** Most significant findings of the qualitative analysis.

Dimension	Categories	Subcategories		Geographical Location and Sex	Disocurses
Identification on social networks	Use of social networks by males and females	Facebook, WhatsApp, Instagram = guys Blogs, Youtube = girls	Use of personal and intimate spaces in relation to other spaces or social media tools.	UF16	We are more about telling each other through WhatsApp, between us or writing to our boyfriend, and of generally posting on Instagram or Facebook, everything goes on there, ah and of giving little away [UF16]
				RF 16	In a blog you don’t have the problem that somebody is there who doesn’t want to be, it gives more privacy than other social networks [RF16]
				RM15	I don’t usually entertain myself at home, I like to go out, I am less interested in social networks, spaces where you have to write, I am more about Instagram and WhatsApp [RM15]
		Productive use= guys Productive use= girls	Motives for using them and the way in which they are used girls and guys.	RM14	Girls often set up a false profile to see what other girls are doing, girls who are not their friends and for whatever reason they do not send a friend request [RM14]
				RF15	The gossip is more our interest, the younger ones don’t care about that stuff [RF15]
			Use of social networks to engage in gossip. Different Female-Male.	RM16	I have sometimes thought of looking at other girls to see what they are wearing or who they are hanging out with, I don’t know if I know more about them, it’s a way of getting closer to someone you like or whatever and them not discovering that you’re following them [RM16]
				RF15	The gossip is more our interest, the younger ones don’t care about that stuff [RF15]
			Expression of feelings in social networks.	RM15	I don’t want them to see me suffer, whilst they could share my joys, I already celebrate that with my friends, but a photo shows how happy I am and makes me happy [RM15]
			Images relating to their body and aesthetic beauty.	UF14	Yes, we care more about image in the photographs we post, maybe because of what they might say, about our image, and you notice that on the networks, we don’t just post anything [UF14]
				UM16	We don’t really ever post photos or anything, or sometimes when we are partying but less of us ourselves, we like to be seen to be having a good time [UM16]
			Difference between the sexes in the degree of intimacy shown by the photographs displayed on social networks.	RF16	Its not only for them that we get pretty, we can also have relationships between us girls, its not just about being a couple, we also like to be pretty for other girls [RF16]
				UF16	Well, sometimes you send photos to attract attention, usually from people you like, to flirt, sometimes I admit that I’ve gone too far, but for now I’ve been lucky and have not had any negative repercussions from them [UF16]
				UM16	The image isn’t so important, man, let’s see, you’re not going to look a mess, but if you show muscle that’s cool, but maybe you don’t have to show the face or how you’re dressed, that doesn’t matter, you can go crappy and nothing happens [UM16]
			Body image.	RM15	Above all I take selfies with my friends, I like to see myself with them and know they are there [RM15]
				UF16	A good selfie saves your life, when you’re down you look at it, send it to your pals, or just to flirt with someone in a chill sense [UF16]
				UM14	As a boy I consider what I wear, how I appear in photos and how I carry myself, looking good doesn’t have to be just for girls, it’s also for us [UM14]
			Emotional expression on social networks	RM14	Boys are not really into talking about our things, about our problems, and less about girls, that would be ridiculous [RM14]
				RF16	Boys are mute, you ask them how they are, if they have something going on or anything, and they don’t answer you, they stay quiet, it seems that nothing ever happens to them, but that’s not the case [RF16]
				UF16	But we like that the guys also share with their issues with us, and more if we are in a relationship, in the way that we girls do, and that doesn’t make them lesser men [UF16]
				UM14	We like send photos more, with friends or to meet up, but not to write, from time to time maybe something a bit affectionate, but to look good [UM14]
			Expressing feelings on social network	RF15	In networks I don’t usually post many things that happen to me, that’s why I use WhatsApp, it’s more intimate, especially with friends [RF15]
				UF16	Always keep in mind the people who may be seeing you, it is better to talk about your things in private, especially if you end up getting it wrong, I do not want people to see me get it wrong [UF16]

## Data Availability

The data is held by the research team and will not be published due to data protection law, but can be consulted on request.

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
