# Peer review of "Digital Identities of Young People from the South of Spain: An Online Sexual Differentiation"

_children, 2021, doi:10.3390/children8121179_

Round 1

Reviewer 1 Report

I would like to congratulate the research team of this paper, the subject of the paper is very opportune and has a high methodological quality. I would also like to thank you for the opportunity to evaluate this excellent work, it is an honour for me. I would like to contribute some points of view that I think can be considered before publication.

In line 131, section "instruments" reference is made to the social network "Tuenti", this social network was closed in 2013 and I think it has been included by mistake, it would be appropriate not to make reference to it.

The study is expandable and there are aspects that are not addressed, although this is justifiable. However, a more extensive "limitations of the study" section indicating some aspects that are not reflected in the study would be appreciated, I indicate some of them.

- Preference of image over video? Creation of more elaborate videos.
- Contemplating homosexuality among the target audience.
- Contemplate the fear of buying or if they have suffered from it.
-...

In the abstract you state "Taking into account these results, we could intervene in the child population to carry out prevention activities focused on social networks" but in the discussion there is no allusion to any type of possible educational proposal that focuses on the results of the study, nor is there any reference to the education received in schools and how these results are linked to the school reality.

We understand that the limitations of the publication do not allow it, but it would be appreciated to see the measurement instruments as an appendix.

Congratulations on this excellent work.

Author Response

We would like to express our gratitude for the time taken to review this manuscript and for the comments made, which we believe to be critical for producing rigorous and quality research. We have detailed below the changes made in the original article: "Digital identities of young people from the south of Spain. An online sexual differentiation"

Modifications have been made in the original manuscript following the reviewers’ comments. For each modification we have written: the original comment as written by the reviewer in addition to the page and line number; and the change made in response to that comment.

REVIEW1

I would like to congratulate the research team of this paper, the subject of the paper is very opportune and has a high methodological quality. I would also like to thank you for the opportunity to evaluate this excellent work, it is an honour for me. I would like to contribute some points of view that I think can be considered before publication.

In line 131, section "instruments" reference is made to the social network "Tuenti", this social network was closed in 2013 and I think it has been included by mistake, it would be appropriate not to make reference to it.

The study is expandable and there are aspects that are not addressed, although this is justifiable. However, a more extensive "limitations of the study" section indicating some aspects that are not reflected in the study would be appreciated, I indicate some of them.

Preference of image over video? Creation of more elaborate videos.

Contemplating homosexuality among the target audience.

Contemplate the fear of buying or if they have suffered from it.

In the abstract you state "Taking into account these results, we could intervene in the child population to carry out prevention activities focused on social networks" but in the discussion there is no allusion to any type of possible educational proposal that focuses on the results of the study, nor is there any reference to the education received in schools and how these results are linked to the school reality.

We understand that the limitations of the publication do not allow it, but it would be appreciated to see the measurement instruments as an appendix.

Congratulations on this excellent work.

ANSWER REVIEW 1

Thank you very much for your kind words that have given us much happiness.

As described we have removed the social network Tuenti in the content of the article (136). It was indeed included with the aim of making a description of all the existing social networks in that age group and we did not notice that it was obsolete. Thank you very much for your appreciation.

We have included a specific section of limitations of the study (section 5) to help the interpretations of the reader as suggested, thank you for your appreciation (507-517)

For the best reading comprehension of the paper, we have included in the introduction section a paragraph that addresses the idea of being able to work with the child population in preventive matters with social networks (78-83) and we have reinforced the idea in the discussion (499-505)

*As requested by the 2 scales from which the attributes of this study are extracted are:

(a) Sex Role Inventory (BSRI), (Bem, 1974);

  1. b) Attributes associated with gender, validated in a study of the Spanish adolescent population "Strong as a dad? Sensitive as a mom?" (Megías and Rodríguez, 2015).

Reviewer 2 Report

Although the paper is dedicated to a very important topic and the mixed-method gives hope that the results will be of great scientific importance, the manuscript does not have an appropriate scientific structure. It is difficult to follow the idea of ​​the authors, it remains unclear why they compared the data in the quantitative part of the research and why only very basic data on the use of SNS was used. The connection between the quantitative and the qualitative part is not congruent. The manuscript should be systematically rearranged to make it clearer why they decided to follow this data and rewritten to make it clear what the author's basic idea is. Research based on the idea of ​​triangulation provides the possibility of very significant scientific insights and the authors should be commended for deciding to do the same. However, in order for the manuscript to be published, I believe that the whole manuscript should be systematically revised,  and clearly present the theoretical basis, problems and hypotheses, and research questions and accordingly clarify the methodological part, as well as the approach to analysis, especially qualitative analysis, and organize both the results and the discussion accordingly. Unfortunately, despite the potential of the data from this research

Author Response

Dear reviewer

We would like to express our gratitude for the time taken to review this manuscript and for the comments made, which we believe to be critical for producing rigorous and quality research. We have detailed below the changes made in the original article: "Digital identities of young people from the south of Spain. An online sexual differentiation"

Modifications have been made in the original manuscript following the reviewers’ comments. For each modification we have written: the original comment as written by the reviewer in addition to the page and line number; and the change made in response to that comment.

REVIEW2

Although the paper is dedicated to a very important topic and the mixed-method gives hope that the results will be of great scientific importance, the manuscript does not have an appropriate scientific structure. It is difficult to follow the idea of the authors, it remains unclear why they compared the data in the quantitative part of the research and why only very basic data on the use of SNS was used. The connection between the quantitative and the qualitative part is not congruent. The manuscript should be systematically rearranged to make it clearer why they decided to follow this data and rewritten to make it clear what the author's basic idea is. Research based on the idea of triangulation provides the possibility of very significant scientific insights and the authors should be commended for deciding to do the same. However, in order for the manuscript to be published, I believe that the whole manuscript should be systematically revised,  and clearly present the theoretical basis, problems and hypotheses, and research questions and accordingly clarify the methodological part, as well as the approach to analysis, especially qualitative analysis, and organize both the results and the discussion accordingly. Unfortunately, despite the potential of the data from this research

ANSWER REVIEW 2

Thank you very much for your comments regarding our manuscript, once analyzed everything it suggests I can only thank your pertinent advice that have helped us to make modifications that have helped to increase the scientific quality of the paper.

We have modified each of the sections of the manuscript, from the introduction to the conclusions. The methodology has been revised to include nuances, explanations and even references that lend themselves to greater understanding by the reader. We have included in detail the methodological strategy and its procedure to ensure replicability and deepened in the qualitative methodology section.

On the other hand, we have substantially modified the results to facilitate their understanding. As well as the discussion.

Reviewer 3 Report

This paper is timely and important and one much needed int he field. the material is important and exciting.

HOWEVER: the paper needs some more work.

first there are superficial but annoying language mistakes and overall some grammar and syntax issues need to be improved.

srcond we need some more information as to how and why discourse analysis was applied on the interviews. How do we see that? what does it add?

the findings are fascinating but are presented in the form of a report. we need more typological care, more analysis more discussion, perhaps even more references to existing lit and lit gaps.

since discussing gender: please use gender appropriate language, boys might come before girls but male does not come before female etc.

Author Response

Dear the editor and reviewer

We would like to express our gratitude for the time taken to review this manuscript and for the comments made, which we believe to be critical for producing rigorous and quality research. We have detailed below the changes made in the original article: "Digital identities of young people from the south of Spain. An online sexual differentiation"

Modifications have been made in the original manuscript following the reviewers’ comments. For each modification we have written: the original comment as written by the reviewer in addition to the page and line number; and the change made in response to that comment.

REVIEW3

This paper is timely and important and one much needed int he field. the material is important and exciting. HOWEVER: the paper needs some more work.

  1. first there are superficial but annoying language mistakes and overall some grammar and syntax issues need to be improved.

Thank you for this appreciation, we have asked for help from a translator to help us in grammatical errors

  1. srcond we need some more information as to how and why discourse analysis was applied on the interviews. How do we see that? what does it add?

Very pertinent this comment, we have included in the article the application of discourse analysis to improve reader understanding, as well as expanded and improved the methodology section. Very grateful for your appreciation.

  1. The findings are fascinating but are presented in the form of a report. we need more typological care, more analysis more discussion, perhaps even more references to existing lit and lit gaps.

We have modified the results section substantially, as well as the presentation of them to improve the scientific quality of the paper. We have completed the discussion to ensure the internal coherence of it. We have also added some references both in the introduction and in the discussion that increase the attractiveness of the article.

  1. Since discussing gender: please use gender appropriate language, boys might come before girls but male does not come before female etc.

Thank you very much for this very accurate assessment, we have been able to modify it in the content of the manuscript - we have observed that this error has been made on numerous occasions, you can see its correction in each of the sections of the manuscript - and we apologize for the use of gender appropriate language on these occasions.

Round 2

Reviewer 2 Report

This version of the manuscript has been greatly improved. However, as the authors deal with issues of gender stereotypes and the use of the Internet, I think it would be important in the introductory part to look at the functioning of stereotypes, especially those related to gender, theories that explain the use of media. Consider whether it makes sense to consider the results in the context of stereotype threat as well. I still believe that there is room for clearer identification of research issues. It is unclear to me why the topics are separated in different tables in the qualitative analysis, I suggest adding a table summarizing which all important topics will be considered. Also, check in Table 3 because the other names are in Spanish (chicos). With additional changes, publication of the paper may be considered.

Author Response

Thank you very much for your very pertinent words and comments.

As he suggests, we have incorporated into the introduction references and a paragraph that analyze the functioning of stereotypes, especially those related to gender, theories that explain the use of the media. Line 79-82

On the other hand, as indicated, we have generated a new table, nº13 (line 413), in which we have merged those most relevant results according to categories and subcategories that we believe can facilitate reading comprehension and improve the scientific quality of the paper.

Thank you very much for your time and your speed of response.

An affectionate greeting
